# Autonomous Vehicles and Vulnerable Road-Users—Important Considerations and Requirements Based on Crash Data from Two Countries

**DOI:** 10.3390/bs11070101

**Published:** 2021-07-15

**Authors:** Andrew Paul Morris, Narelle Haworth, Ashleigh Filtness, Daryl-Palma Asongu Nguatem, Laurie Brown, Andry Rakotonirainy, Sebastien Glaser

**Affiliations:** 1Transport Safety Research Centre, Loughborough University, Loughborough LE11 3TU, UK; a.j.filtness@lboro.ac.uk (A.F.); daryl-palmaasongu.nguatem@student.uhasselt.be (D.-P.A.N.); l.a.brown2@lboro.ac.uk (L.B.); 2Centre for Accident Research and Road Safety, Queensland University of Technology, Queensland (CARS-Q), Brisbane 4000, Australia; n.haworth@qut.edu.au (N.H.); r.andry@qut.edu.au (A.R.); sebastien.glaser@qut.edu.au (S.G.)

**Keywords:** vulnerable road-user, connected and autonomous vehicles, road intersections, crash data

## Abstract

(1) Background: Passenger vehicles equipped with advanced driver-assistance system (ADAS) functionalities are becoming more prevalent within vehicle fleets. However, the full effects of offering such systems, which may allow for drivers to become less than 100% engaged with the task of driving, may have detrimental impacts on other road-users, particularly vulnerable road-users, for a variety of reasons. (2) Crash data were analysed in two countries (Great Britain and Australia) to examine some challenging traffic scenarios that are prevalent in both countries and represent scenarios in which future connected and autonomous vehicles may be challenged in terms of safe manoeuvring. (3) Road intersections are currently very common locations for vulnerable road-user accidents; traffic flows and road-user behaviours at intersections can be unpredictable, with many vehicles behaving inconsistently (e.g., red-light running and failure to stop or give way), and many vulnerable road-users taking unforeseen risks. (4) Conclusions: The challenges of unpredictable vulnerable road-user behaviour at intersections (including road-users violating traffic or safe-crossing signals, or taking other risks) combined with the lack of knowledge of CAV responses to intersection rules, could be problematic. This could be further compounded by changes to nonverbal communication that currently exist between road-users, which could become more challenging once CAVs become more widespread.

## 1. Introduction

Motor vehicles are becoming increasingly advanced, and many functions that support drivers in various traffic situations (e.g., adaptive cruise control, forward-collision warning, and pedestrian safety) are already on the market. It is expected that this trend will continue, and that future connected and autonomous vehicles (CAVs) will be capable of handling most of the manoeuvring and control functions of the vehicle in all traffic scenarios [1]. In urban settings, this means that CAVs are required to interact with pedestrians, motorcyclists, and cyclists. In this study, pedestrians, motorcyclists, and cyclists were considered to belong to a classification of road-users known as vulnerable road-users (VRUs), so called because they are more vulnerable to injury if road collisions occur.

Bicycles can be quickly avoided, needing little space and providing sparse early cues about their intended manoeuvres. Consequently, the complexity of the driving task increases, and drivers (or CAVs) who are only expecting typical motor-vehicle movements may fail to detect and appropriately respond to these manoeuvres [2]. This unpredictability provides a great challenge for human drivers, and there are not adequate data to determine whether CAVs face a similar challenge. It is certainly important for CAV systems to be specifically trained to respond to these behaviours. Motorcycles are very appealing to road-users, because their compact size aids riders in easily moving in and out of traffic. However, their size also has several disadvantages. For example, they are lightweight, which means that riders could easily lose control on uneven road surfaces, an object on the road, or inadequately placed street furniture [3,4]. Pedestrians are distinguished by several key features, such as personal choice, variable dynamics, and vulnerability. Debatably, they have the least predictable road-user behaviour characteristics, and differences can be influenced by features such as walking speed, age, knowledge of the environment, individual or group transit, and time of day [5,6,7].

These differences in VRU behaviour require CAVs to reliably detect other such road-users, but also require VRUs to interact with CAVs that are equipped with different levels of automation. Hence, this necessitates different response requirements in different traffic situations and circumstances for such interactions to happen safely [8].

Although it is not yet certain when fully autonomous vehicles will be functional on roads, if ever, there is some speculation that, by 2040, most fleets will be at least semi-autonomous [9]. As such, pedestrians, cyclists, motorcyclists, and CAVs must learn to coexist.

Research into human behaviour relating to CAVs, including user acceptance, is gaining interest [10]. This is an important area of study; in many countries, studies have been conducted to assess public opinion on CAVs, for example, in Australia [11], the United Kingdom [12], and elsewhere across the world [13,14,15]. Findings generally indicate that users see benefits in CAVs, but there are concerns regarding safety, data security, lack of openness from developers, etc. Studies have also showed that the perceived value of CAVs influences a user’s trust, and that trust tends to increase the longer that a person is exposed to the CAV. However, although some studies consider the perspectives of other road-users such as VRUs, the majority are concerned with studying drivers’ attitudes to adopting CAVs, and very few studies have asked drivers or VRUs to predict exactly how their behaviours might change if they were to use or interact with a CAV. Furthermore, there is a lack of research that compares the views of road-users in different countries.

This developing research is further reflected in an increasing number of research programmes within this field, including the Automated Vehicle Research Programme of the U.S. Department of Transport, and the Human Factors in Automatic Driving project involving a consortium of European research institutes and car manufacturers. Both programmes, however, specialise in the motive force and not on (the perspective of) other road-users, including VRUs [1]. Human error is attributed as the largest contributor to crashes [16]; therefore, researchers and road-safety experts emphasise that the potential road-safety benefits of CAVs could be achieved by relieving humans from the responsibility of driving. CAVs do not make human errors and do not deliberately violate traffic regulations; therefore, they are assumed to outperform human drivers, and thus contribute to a substantial reduction in road collisions [17,18,19]. However, some researchers express certain reservations about such expectations. There are also uncertainties associated with the interaction of CAVs with nonautomated road-users, particularly VRUs; therefore, subsequent safety effects on this group of road-users create cause for concern [6,20]. To date, research on the interactions between CAVs and VRUs has been limited to the technical aspects of the detection and recognition of pedestrians, motorcyclists, and cyclists by vehicles, again solely considered from the attitude of the vehicle [1]. However, the authors in [1] also noted that it is equally important to look at matters from the attitude of the VRUs. Are VRUs able to effectively interact with CAVs? As an example, would this affect their crossing decisions or their red-light compliance? If so, how? Would they accept smaller gaps or would they prefer larger safety margins? Would they be inclined to infringe traffic controls such as red lights more often or not? Through the transitional period, with its combination of fully or partly autonomous and manually driven vehicles, are pedestrians capable of differentiating between these vehicles, and would they accordingly adjust their behaviour? Furthermore, during the transition, there is likely to be a large fleet of partially autonomous vehicles on the road, and these vehicles may suddenly hand back control to the human driver if they encounter a situation for which they are not programmed to handle. Much research has been dedicated to understanding the safety concerns around this issue, in particular, whether the human driver could have the appropriate situational awareness to react to a situation in time. Returning control to a human driver who is not ready to re-engage with the driving task could have particularly severe consequences for VRUs.

Studies on the interaction between CAVs and VRUs, and answers to the above questions, have received relatively limited attention [6,21]. Therefore, it is difficult to both estimate the safety effects of a transition towards automated vehicles and identify the actions to minimize the risk that interactions between CAVs and non-automated road-users cause, and whether they induce unsafe situations and accidents [1].

Therefore, this study will evaluate existing real-world collision data relating to the extent of vulnerable road-user collisions as a proportion of all crashes, across two countries. This approach will highlight possible scenarios which are likely to be problematic for this road-user group in future interactions with CAVs. By using comparable methodological approaches, for the first time, results can be compared between countries, giving great insights into the potential problematic interactions that future CAVs and VRUs must overcome. Overall, the following research questions are addressed:How prevalent were VRU accidents within current road collision statistics in the period of 2017–2019? (Due to data availability, specifically within Great Britain and Australia);What are currently the most common scenarios for collisions with VRUs as shown in the crash data from both countries?;What are the potential advantages of the introduction of CAVs for VRUs?;What are the future data requirements to monitor the effects of CAVs on VRUs?;What recommendations could be made from this study?

This study adopts a harmonised approach to CAV development that considers variations in traffic scenarios between different countries and how the protection of VRUs can be realised in an increasingly connected road transport system. The study focuses on the traffic situation in two different regions of the world (Great Britain and Australia), and in each country, the progress towards a partial fleet of CAVs is thought to be comparatively rapid. However, the numbers of VRUs and the frequencies and methods of utilising the road system are thought to differ considerably between these countries. Therefore, a further rationale for the study is to consider how CAV technology can be universally applied in relatively heterogenous traffic environments.

## 2. Methods

National and state collision data were analysed in the 2 countries to examine the extent of the VRU casualty problem and to consider scenarios and situations in which VRUs are injured and killed in traffic accidents. In both Great Britain and Australia, cyclists are allowed on many roads but are forbidden to use motorways and highways. In some towns and cities in both countries, dedicated cycle lanes are provided, and these are growing in number as local authorities encourage sustainable transport. Pedestrians are not conventional road-users, and usually only encroach on the road when they are seeking to cross over from one side to the other. Additionally, pedestrians are sometimes necessarily present on the roads when dedicated pavements are not provided, particularly in rural areas, where they have no option but to walk on or close to the roadway.

### 2.1. Great Britain Data

In Great Britain, the national collision database that was used to assess the extent of the casualty problem involving VRUs was the STATS19 database (using data from the year 2019). STATS19 is a national road collision database which is founded on collision records completed by police officers in the event of a collision occurring in Great Britain. To be included as a collision record within the STATS19 database, the collision must be reported to the police and should involve human injury or death. The STATS19 data collection form collects a wide variety of information about the collision (such as time, date, location, and road conditions) together with the vehicles and casualties involved and contributory factors to the collision (as interpreted by the police officer). The form is completed at either the scene of the collision, or when the collision is reported to the police. The STATS19 data are collected every year and overview analysis is provided below, as was presented in the U.K. Department for Transport Statistical Release [22]. STATS19 data were also analysed in this study.

### 2.2. Australian Data

The Australian Road Deaths Database is compiled by the Bureau of Infrastructure, Transport and Regional Economics (BITRE) [23], and contains basic details of road transport collision fatalities in Australia. A report on fatalities is produced each month. BITRE also publish summary data on hospitalised injuries from road collisions which are produced by the National Injury Surveillance Unit at Flinders University under an agreement with the Australian Institute of Health and Welfare. Full-year fatality data were available up to and including 2019, but hospitalised injury data were only available up to and including 2017 at the time of writing. Details of collision types and locations are not available at the national level, but are compiled by state and territory transport departments from information provided by the state police forces. These individual databases vary in reporting requirements, time lags, severity coding, and the variables collected.

Data from both countries were analysed using the statistical analysis software package SPSS.

## 3. Results

### 3.1. Vulnerable Road-Users in Great Britain

Considering the three vulnerable road-user groups identified previously, in Great Britain during 2019, 54,905 VRUs were injured in road collisions (36% of all road collision casualties). A total of 895 VRUs were fatally injured in collisions (comprising 52% of all fatally injured road-users), and a total of 16,684 road-users were killed/seriously injured (KSI—comprising 56.4% of all KSI road-users, [24]). These are summarised below in Table 1.

In the majority of collisions involving other vehicles, the VRU was involved in a collision with a passenger car (~70% of all VRU casualties), with the effect being highest for cyclists (77% of casualties) and pedestrians (75% of casualties), whereas 62% of motorcyclist casualties were involved in collisions with passenger cars.

Absolute numbers of casualties as described above are important, but they do not paint a complete picture because they do not consider exposure factors such as the total distance travelled. If the VRU casualties are examined from the perspective of one billion passenger miles travelled [24], analysis shows the overwhelming challenges that this road-user group presents within a safety context. The fatality rate per billion passenger kilometres for pedestrians is nearly 19-fold higher than the corresponding rates for passenger cars, and 85-fold higher than for large goods vehicles (e.g., trucks). This analysis is shown in Table 2.

It is also important to examine the nature and circumstances of VRU collisions and examine the key “scenarios” in Great Britain for such collisions. Table 3 shows the percentage of collisions occurring at (or not at) intersections in accidents of all injury severities. As can be seen in Table 3, all VRU collisions are slightly more prevalent at road intersections compared to collisions not at intersections, particularly for cyclist accidents. When considering only accidents where the casualty was killed or seriously injured, the percentages were similar, although cyclists and motorcyclists showed a slightly higher proportion not occurring at intersections (37% and 46%, respectively, compared with 32% and 38% when all injury severities are considered).

The term “intersection” includes several types of road intersections, including signalised and non-signalised road intersections, roundabouts, etc. (although the data do not satisfactorily discriminate between signalised and non-signalised intersections). However, just under half of pedestrian collisions occur at locations remote from intersections, and these are thought to involve pedestrians crossing at random points on the roads away from designated crossing facilities. Non-signalised intersections present unique challenges for CAVs because they often require decision-making and predictions about safe manoeuvring in relatively congested traffic environments.

It is interesting to further examine the breakdown of collisions according to intersection type and road-user type for all collision severities. As shown in Table 3, T-intersections appear to be the most common intersection types and typically involve collisions in which: (1) the VRU attempts to join live traffic lanes but must cross potentially fast-moving traffic (prevalent for cyclists and motorcyclists); or (2) the collision partner attempts to join live traffic lanes but fails to spot an oncoming VRU (prevalent for motorcyclists). Again, this provides an indication of the types of manoeuvres that CAVs will need to perform to safely merge in traffic, and indicates the importance of vehicle sensing systems and critical decision-making algorithms that will be required to deal safely and effectively with VRUs. The important issue of pedestrians crossing remotely from intersections is further highlighted in these data.

### 3.2. Vulnerable Road-Users in Australia

In Australia, during 2019, 18,521 VRUs were admitted to hospital following road accidents (47% of all road collision hospitalisations). A total of 410 VRUs were fatally injured in accidents (comprising 34% of all fatally injured road-users) [23]. These data are presented below in Table 4.

Those VRU crashes which involve motor vehicles are of most relevance in relation to understanding the safety of autonomous vehicles interacting with VRUs. Although these crashes comprise a large majority of the VRU road fatalities, they comprised only 23% of cyclist and 29% of motorcyclist hospital admissions in 2013–2017. Collisions with motor vehicles comprised 90% of pedestrian hospital admissions because falls (conceptually “single-vehicle pedestrian crashes”) are excluded from the road collision data, unlike falling from a bicycle or a motorcycle which are included. In the majority of the serious injury crashes involving other vehicles, the VRU was involved in a collision with a passenger car (~83% of all VRU casualties), with the effect being highest for pedestrians (90% of casualties), 83% of motorcyclist casualties, and then cyclists (75% of casualties).

The vehicle occupant fatality rate was 3.0 per billion four-wheeled motor vehicle-kilometres travelled in Australia in 2019, whereas the relevant motorcyclist fatality rate was 86.4, almost 30 times higher [23]. The absence of comprehensive travel surveys has resulted in fatality and casualty rates for non-motorised VRUs being unknown in Australia. In the absence of national data, [25] attempted to estimate the relative risk of cycling compared to driving. They concluded that the fatality risk for cycling was about 5 to 19 times greater than driving (based on data from Melbourne and Sydney). The estimated relative risk of injury varied greatly on the data source, although appeared to be higher than that for fatalities.

Information regarding the locations and collision types for non-fatal crashes is not available at the national level in Australia. Therefore, the tables below present data from the state of Queensland, which has a wide mix of urban and rural travel and may be an appropriate representation of the patterns in national data. However, the data include only those VRU crashes reported to police; therefore, the total numbers are likely to be substantially underestimated.

Table 5 shows that, in Queensland, about three-quarters of casualties from cyclist-motor vehicle crashes occurred at intersections, compared to about half of the pedestrian–motor vehicle and motorcycle–motor vehicle casualties. In Queensland, adults are legally allowed to ride on footpaths, and this can contribute to crashes when cyclists are moving from a driveway to the road or motor vehicles are entering a driveway. Driveways were included in the definition of intersections used here. Our earlier study [2] reported that more than 60% of fatal and serious injury bicycle–motor vehicle collisions across Victoria, Queensland, and South Australia in 2012–2015 occurred at intersections, with the percentage being somewhat higher in Queensland. The lower percentage of motorcyclist collisions at intersections in Queensland compared with Great Britain may reflect the greater prevalence in Australia of recreational riding (often on country roads with few intersections) compared to commuter riding. When considering only accidents where the casualty was killed or seriously injured, similar results were seen, although there was a slightly higher proportion of pedestrian incidents not occurring at intersections (57% compared with 53% when all injury severities were considered).

Table 5 additionally shows the type of intersection at which the crash occurred. For each type of VRU, crashes were most common at T-intersections. Motor vehicle collisions involving pedestrians and motorcyclists were more common at crossroads than at roundabouts, but similar numbers of cyclist–motor vehicle collisions occurred at crossroads and roundabouts. Although the prevalence of these types of intersections in the areas in which the three types of VRUs travel may explain much of the difference in the types of intersections in collisions, there is evidence that roundabouts do not confer the same safety benefits to cyclists that they do to motor vehicle users [26,27,28]. Given the potential for VRUs to take a range of trajectories through roundabouts, it may be more difficult for AVs to detect, predict, and respond safely to VRUs in roundabouts than in other intersection types.

Our earlier study identified that the proportion of different bicycle collision types in collisions with motor vehicles differed across states. For example, “From footway-manoeuvring” crashes (which do not include crashes with vehicles emerging from driveways) accounted for almost one-quarter of all bicycle–motor vehicles crashes in Queensland, but fewer than 10% of these crashes occurred in Victoria (see Table 6).

## 4. Discussion

In both Great Britain and Australia, vulnerable road-users (VRUs) comprise a relatively high proportion of injured road-users. In Great Britain, VRUs comprised over 50% of fatally injured road-users during 2019, whereas during the same year in Australia, the number of VRU fatalities was slightly lower (34%). Despite these differences, this road-user group represents a challenge for national road authorities and policymakers because the opportunities for injury prevention are limited by the VRU exposure to collision forces when they are involved in crashes with road vehicles including passenger cars, trucks and buses. In Great Britain, over 50% of each VRU type (i.e., pedestrian, cyclist and motorcyclist) are seriously injured/killed at road intersections, and although national data are not available in Australia, data from the state of Queensland suggest a similar story, although the numbers of seriously injured/killed cyclists are slightly lower than in Great Britain, which may be due to differences in cycling provisions. Nevertheless, these data are indicative of the types of scenarios that future connected and autonomous vehicles (CAVs) may face as they become more prevalent in future. Some potential challenges that CAVs may have to deal with also include the fact that: (1) CAVs will need to demonstrate a capability to deal with unpredictable actions that VRUs may take at intersections. Examples of these include distracted pedestrians (e.g., using mobile phones and wearing headphones) stepping out into free-flowing traffic and both cyclists and motorcyclists manoeuvring in vehicle blind spots, although many more examples are evident; and (2) VRUs will need to understand the capabilities and limitations of CAVs so that the two road-user groups can coexist. What are the implications of these challenges? The current study used macroscopic data to understand the problems at the top-level; therefore, the answer to this question requires further investigation. This will likely involve additional analyses which are able to probe more detailed data (involving, for example, data from in-depth crash investigations). Such analysis will also need to be undertaken with a full understanding of the capabilities and limitations of CAV technologies so that the more challenging scenarios (sometimes known as “edge-cases”) can be identified with a view to ensuring that the technology will adequately and reliably deal with them.

At present, VRUs can sometimes predict the likely actions of vehicles on the basis of speed and traffic flow, while vehicle intentions can be communicated to a certain extent to the VRUs through an implicit code of conduct. When CAVs become prevalent, this dynamic will change and may even cease to exist. Much research is being carried out to understand this issue [29,30,31,32], and CAV manufacturers are using this learning to develop potential solutions, providing methods for the vehicle to signal its intent to other road-users. However, it is very unlikely in the short term that this will exactly mirror the types of communication VRUs are used to receiving from human drivers. VRUs must adapt and learn new methods of communication, and, more critically, a VRU must quickly recognise whether they are dealing with a human, or one type of CAV or another. This is so they can develop capabilities to convey these essential communications in time to prevent a hazardous situation from occurring.

When discussing how CAVs will handle the unpredictable actions of VRUs, it is important to consider the current and expected capabilities of CAVs, as well as their potential limitations. Most CAVs use a range of sensors for object detection, the most common being RADAR, LiDAR, cameras, and ultrasonic. Much research has been carried out that describes the capabilities of each of these [33,34]; more recently, many studies have focused on so-called “sensor fusion”, whereby the benefits of using different combinations of sensors are explored [35,36]. Furthermore, cybersecurity is a crucial concern in CAV development, and it has been shown that, even if a sensor array could perform very well, it can still be vulnerable to attack from outside sources, and such attacks could have devastating consequences [37]. Research also shows that some sensors will suffer the same issues as humans when faced with obscuration, for example, due to weather conditions (fog, heavy rain, etc.).

Of particular interest to the current study is how these sensors perform in relation to VRU detection. The authors of [38] recently conducted a review on this subject. They found that although a comprehensive suite of sensors would achieve high levels of detection, they can still be limited by the algorithms which interpret the data from the sensors, as well as a lack of processing power to process these data quickly enough. Additionally, [39] conducted an extensive review of the typical architecture of self-driving vehicles, highlighting the requirements of both the “perception system” and the “decision-making system”. Therefore, in addition to pure detection capabilities, there are also the CAVs’ decision-making algorithms to consider. Humans take lessons before driving alone, and it is accepted that novice drivers tend to have a higher accident rate than more experienced drivers, indicating that experience is correlated with a driver’s ability to drive safely. How will CAVs gain this experience? Between January and December 2019, Waymo vehicles in the United States (with trained safety drivers on board) drove 6.1 million miles in and around U.S. urban areas. In addition, from January 2019 until September 2020, the fully driverless versions drove 65,000 miles, and collectively, these two driving scenarios were claimed to equate to approximately 500 years’ worth of human driving. During this period, Waymo vehicles were involved in 47 contact events with other road-users, including other vehicles, pedestrians, and cyclists. Nearly all collisions were the fault of a human driver or pedestrian, and none resulted in any “severe or life-threatening injuries”. Although the assertions do not consider exposure factors including “near-miss events”, these results suggest that CAVs could positively impact pedestrian crashes in urban areas, providing sensor limitations can be overcome. Nevertheless, the Waymo studies were conducted in the United States, which has different road and traffic conditions from those of the United Kingdom and Australia; therefore, it is important to study the transferability of knowledge across different countries as the technology becomes more prevalent.

### 4.1. Implications for Autonomous Vehicles—Pedestrians

Data from Great Britain suggest that around half of pedestrian accidents occur at sites remote from crossing facilities, with many occurring when parked vehicles obscure driver vision. The challenges of younger pedestrians appearing suddenly due to crossing the road while being masked by stationary vehicles, failing to look properly, or being careless and less aware while playing dangerously on the street puts younger people more at risk of being involved in accidents. In contrast, older pedestrians tend to move more slowly on the roads and are more likely to be less able to judge the path and speed of a vehicle. They are also commonly recorded as having contributory factors related to their wellbeing, such as mental or physical illness [6,7]. Many predict that advances in automated vehicle technology already available on CAVs will reduce pedestrian fatalities substantially through eliminating crashes caused by human error, including late braking, distraction/inattention, secondary non-driving tasks (e.g., mobile phone usage) and missed observations. However, vehicle sensing capabilities will be important—one study [40] commented that sensors’ abilities to detect pedestrians in advance of fatal collisions vary from <30% to >90% of fatalities. Most research agrees that combining sensor technologies offers the greatest potential for eliminating fatalities, although for this to be feasible, it may require the more expensive sensors to be reduced in cost. Furthermore, whereas the initial deployment of CAVs is likely be restricted to highways and select urban areas, urban streets typically account for a substantial share of pedestrian fatalities [41]. Many pedestrians do cross at designated crossing facilities—in such situations, it could be expected that benefits may well be gained from the introduction of CAVs because they will be much more likely to obey the road rules (e.g., traffic signals, signs, etc.) compared to human drivers, and will be much less prone to human errors (of the types listed above).

Other scenarios in addition to those described could be more challenging. For example, areas around schools, including the campus and surrounding streets, present a difficult design challenge for developers of automated driving systems. The inconsistency of traffic conditions and procedures, an increased density of traffic during peak times, and frequent interactions with schoolchildren pedestrians in school zones presents a difficult design challenge for CAVs. It is imperative that CAV developers understand and address these characteristics before CAVs are deployed in school zones, so appropriate technology, design, and regulatory approaches can be implemented. To date, there has been no coordinated effort to assess the unique challenges of deploying CAVs in school zones or to systematically identify relevant research gaps, and there is little guidance for CAV developers in this respect.

### 4.2. Implications for Autonomous Vehicles—Cyclists

Data from Great Britain suggest that intersections are a major consideration in accidents involving cyclists, with 67% of accidents occurring at these road locations. The authors of [42] argue that CAVs may offer the potential to reduce cycling accidents due to their greater ability to detect and monitor the full range of road-users compared to human drivers, as was described previously. Specific points noted were the eradication of blind spots due to the CAVs’ long-range detection sensors and the future potential capability of bicycles and/or bicycle helmets to communicate with CAVs via transponders. However, the sensing technology issues raised earlier, such as vulnerability to obscuration or insufficient processing power to react quickly, are also evident with this road-user group. Many accidents are thought to occur at intersections when the driver of the road vehicle pulls out of the intersection into the path of the cyclist. As [8] suggest, in such traffic situations in which informal right-of-way rules are applied, road-users often communicate non-verbally to clarify their intentions and ensure a smooth interaction, including eye contact, nodding, and hand gestures, which predict attention and awareness. These mechanisms for interacting in traffic may be of limited (if any) use in situations with automated vehicles, for which developers are designing unique and varied methods of their own for communication. VRUs will need to learn new methods of informal communication that are relevant to the specific CAV they encounter, requiring quick and precise identification of which style of communication is needed. Cyclist expectations of the behaviour of an automated vehicle might also be incorrectly based on the expectations of a manually driven vehicle, or unproven, possibly unrealistic characteristics. For example, what will happen when cyclists blindly assume that self-driving cars will yield and stop for them at intersections and when turning? Furthermore, what will be the effect of making eye contact with the driver (or controller) of an autonomous vehicle, if this person is not the person deciding when to brake? Another important challenge, especially for any transition period, will be to ensure that road-users can distinguish (partly) automated cars from manually driven cars.

### 4.3. Implications for Autonomous Vehicles—Motorcyclists

As with cyclists, data from Great Britain suggest that intersection locations are an important factor in 58% of accidents involving motorcycles. As with cyclist accidents, common scenarios involve vehicles pulling out into path of motorcycles, motorcycles and other vehicles colliding at crossroads (both going ahead and neither vehicle yielding), and other vehicles turning across the path of oncoming motorcyclists. Motorcyclists are arguably the most problematic group because they are the only road-users who share all kinds of road and traffic environment conditions, including the full velocity range, with cars. The Motorcycle Industry in Europe (ACEM) through its position paper [43] argues that available sensing systems on current CAVs may be good at detecting large objects but are not so good at detecting smaller vehicles, such as motorcycles. As observed by [44], for CAVs to become mainstream, the accident-avoidance rate must be far more efficient than that of manual driving. However, because the risk is far greater when motorcyclists are involved (because of the speeds involved), the pressure for effective autonomous collision avoidance increases substantially. With the motorcycles’ smaller frame and superior manoeuvrability, CAV sensors will need to have a wider range so that there are no blind spots where a motorcycle may enter.

The rush to market driver-assist systems and CAVs (both semi-and full-autonomous vehicles) may pose a significant threat to motorcyclists when the developers of this technology and the vehicle manufacturers are not held to the highest safety standards throughout the entire development and implementation process. Evidence does suggest that CAV manufacturers do consider VRUs during the development process; indeed, a number of safety systems are targeted specifically at VRU protection. However, research and CAV safety assurance procedures still provide greater focus on how to navigate alongside cars, truck, etc., and more could be done to ensure that VRUs are a priority. Furthermore, developers must consider variances in VRU behaviour between all countries in which they intend for the CAV to operate. If CAV systems are not conceived and developed with motorcycles and motorcyclists in mind, the eventual result could be that motorcycles are excluded from certain roadways, or worse, banned from roads altogether.

## 5. Conclusions

The current study explored data from Great Britain and Australia. In these two countries, although a different range of road layouts may be evident, VRU behaviours and driver attitudes are generally equal in terms of technology development and the acceptance of new technologies. The main findings from the study are as follows:Vulnerable road-users make up over one-half of fatally injured road-users in Great Britain and over one-third in Australia;Intersection crashes involving VRUs are very common in both countries. Over 50% of crashes occur at an intersection of some type in both countries;Intersection crashes are particularly problematic for motorcyclists and cyclists in Great Britain and motorcyclists in Australia;Crashes at both signalised and non-signalised intersections may diminish when autonomous vehicles become widespread; as such, vehicles are more likely to more stringently adhere to road rules and regulations, thereby giving the VRU more certainty regarding safe crossing opportunities;However, there may still be challenges ahead based on CAVs’ and VRUs’ understanding each other’s codes of conduct on the roads while considering unpredictable behaviour, particularly at intersections;Other non-intersection road locations may also present challenges, and there is scope for understanding and defining the various “edge-case” scenarios where potential problems may manifest.

Further research should also be conducted to understand how the results presented here transfer to countries that are less developed, that would receive great road safety benefits from CAVs but face very different challenges. For example, the work in [45] highlighted issues such as trust, and acceptance is particularly challenging in countries that currently have very little advanced technology within the transport system. The views and behaviours of road-users in less developed countries, particularly VRUs, must be understood by CAV developers to ensure that CAVs are implemented safely and successfully in these areas.

## Figures and Tables

**Table 1 behavsci-11-00101-t001:** Vulnerable road-users in a Great Britain road collision context (2019).

Road-User Type	Fatally Injured (n)	%	Total Casualties (n)	%
Pedestrians	462	26%	21,836	14%
Cyclists	98	6%	16,873	11%
Motorcyclists	335	20%	16,196	11%
**All Road-Users**	**1748**		**153,315**	

**Table 2 behavsci-11-00101-t002:** Casualty and fatality rate by passenger kilometres travelled.

Road-User Type	Casualty Rate per Billion Passenger Kilometres	Fatality Rate per Billion Passenger Kilometres
Pedestrians	2595	55
Cyclists	8154	47
Motorcyclists	8809	182
Passenger cars	339	2.9
Buses/coaches	227	1.0
Large goods vehicles	80	0.6

**Table 3 behavsci-11-00101-t003:** Vulnerable road-user casualties by intersection type.

Road-User Type	Not at Intersection	T-Intersection	Roundabout	Crossroads	Other/Unknown Intersection Type
Pedestrians (*n* = 21,836)	45% (9906)	31.5% (6831)	4% (880)	10% (2182)	9.5% (2037)
Cyclists (*n* = 16,873)	31.5% (5338)	37% (6228)	13.5% (2274)	11.5% (1971)	6.5% 1062)
Motorcyclists (*n* = 16,196)	38% (6112)	33.5% (5436)	9.5% (1499)	10% (1648)	9.5% (1501)

**Table 4 behavsci-11-00101-t004:** Vulnerable road-users in the Australian road collision context (casualties are hospital admissions as recorded by health authorities).

Road-User Type	Fatally Injured—2019 (*n*)	%	Total Hospitalised Casualties—2017 (*n*)	%
Pedestrians	160	13%	2711	7%
Cyclists	39	3%	7077	18%
Motorcyclists	211	18%	8733	22%
All Road-Users	1195		39,330	

**Table 5 behavsci-11-00101-t005:** Vulnerable road-user casualties and intersection types of police-reported crashes involving motor vehicles—Queensland, Australia, 2019.

Road-User Type	Not at Intersection	T-Intersection	Roundabout	Crossroads	Other Intersection Type
Pedestrians (*n* = 648)	52.8% (342)	20.7% (134)	2.9% (19)	14.8% (96)	8.8% (57)
Cyclists (*n* = 769)	25.7% (198)	32.9% (253)	14.6% (112)	13.4% (103)	13.4% (103)
Motorcyclists (*n* = 1585)	47.3% (749)	22.0% (349)	7.9% (126)	11.9% (189)	10.9% (172)

**Table 6 behavsci-11-00101-t006:** Most common fatal and serious injury bicycle–motor vehicle collision manoeuvres in Victoria and Queensland, 2012–2015.

**Victoria**	**Queensland**
**Cash Type**		**n**	**%**	**Crash Type**		**n**	**%**
**Right through-opposite directions**	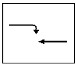	**206**	**17.6**	**From footway-manoeuvring**	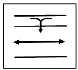	**226**	**24.2**
**Cross traffic–adjacent approaches**	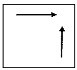	**127**	**10.8**	**Cross traffic-adjacent approaches**	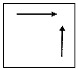	**113**	**12.1**
**Vehicle door-on path**	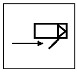	**110**	**9.4**	**Right through-opposite directions**	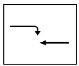	**103**	**11**
**From footway-manoeuvring**	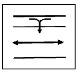	**103**	**8.8**	**Right near-adjacent approaches**	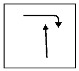	**81**	**8.7**
**Emerging from driveway/lane-manoeuvring**	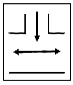	**69**	**5.9**	**Left near–adjacent approaches**	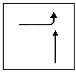	**70**	**7.5**
**Rear end–same direction**	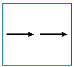	**69**	**5.9**	**Other manoeuvring**		**59**	**6.3**
**Right near–adjacent approaches**	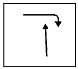	**65**	**5.6**	**Left turn sideswipe–same direction**	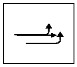	**33**	**3.5**
**Left near–adjacent approaches**	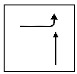	**64**	**5.5**	**Emerging from driveway/lane-manoeuvring**	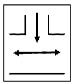	**32**	**3.4**
**All other manoeuvres**		**359**	**30.0**	**All other manoeuvres**		**215**	**23.0**
**Victoria total**		**1172**	**100.0**	**Queensland total**		**952**	**100.0**

## Data Availability

Data for Great Britain can be found online at https://data.gov.uk/dataset/cb7ae6f0-4be6-4935-9277-47e5ce24a11f/road-safety-data (accessed on 30 March 2021).

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
