# Peer review of "Autonomous Vehicles and Vulnerable Road-Users—Important Considerations and Requirements Based on Crash Data from Two Countries"

_behavsci, 2021, doi:10.3390/bs11070101_

Round 1

Reviewer 1 Report

The topic is very actual and important for safe and connected mobility, as a global strategy goal. I thinhk that the authors did a great job. The comments are bellow: 1) Try to define research questions more precisely, so that you will be able to give a precise answer to them in the section "discussion". 2) Try to systematize the data into one or two tables for UK and Australia. 3) The "Discussion" chapter should be designed to provide clear answers to research questions. After the improvement of the paper in line with all comments, I suggest the accepting the paper for publication.

Author Response

RE: [Behavioral Sciences] Manuscript ID: behavsci-1266237 - Revision

Title: “Autonomous Vehicles and Vulnerable Road-Users – important considerations and requirements based on crash data from two countries”

Dear Reviewers,  

On behalf of myself and my co-authors we would like to say a sincere thank you for your time in reviewing our manuscript, we greatly appreciate your feedback and suggestions for improvement. In addressing your comments, we have made substantial changes to the manuscript for which a revised version is attached, and we have also included below our responses to each comment.

We hope you feel that your comments have been addressed satisfactorily and we look forward to hearing any further response.

With very best regards,

The authors

Reviewer 1

Comment

Response

Try to define research questions more precisely, so that you will be able to give a precise answer to them in the section "discussion".

The research questions have been refined and the revised Discussion and (newly added) Conclusions section attempts to provide more precision in the answers to these.

Try to systematize the data into one or two tables for UK and Australia

The tables referring to ‘at/not at intersection’ and ‘breakdown of intersection type’ have been condensed into a single table (for each country).

The "Discussion" chapter should be designed to provide clear answers to research questions.

As above, the Discussion section has been revised.

Reviewer 2 Report

I really like the topic of this study and the approach to compile and discuss crash statistics on VRUs.  This is certainly an important consideration with respect to the performance of CAVs and ADAS applications.  The authors did a very good job setting up this study and explaining the implications of the crash statistics.  However, I think there remains a significant gap here.  I believe it would really help if the authors described the types of sensors commonly deployed on these vehicles to support these systems, such as, cameras, LIDAR, and RADAR.  Highlighting their capabilities, as well as the potential blind spots associated with these systems (mentioned at Line 298 but not described or discussed), including how they differ and could be better or worse than human capabilities, or how they could supplement drivers’ perception in the types of situations noted – would provide valuable context. 

For example, the authors very adeptly note that vehicles pulling out in front of motorcycles is an issue; however, there isn’t a comprehensive description about how this situation could be mitigated or potentially not, by these systems.  There was one related note about a Netherlands study pointing out that CAVs may not perceive motorcycles as well as other vehicles, but that was the only such example (and it was referenced only, not discussed by the authors in the context of the scenario(s) of interest).  Additionally, one could argue that urban areas and intersections with a lot of activity could be handled better with a full sensor array.  And alternatively, it could be described how or why these sensors may still have trouble due to occlusion, limits in on-vehicle data processing capability, or limitations in the ability to differentiate important information from extraneous input ingested through the perception system, etc.

Specifically referenced comments:

Abstract Line 25:  I would be a little careful here indicating that CAVs don’t or will not have the capability to detect non-verbal communication.  For instance, there has already been work to get these vehicles to operate under conditions where they are receiving traffic control instructions from a human in the road (such as a police officer). Though certainly more development is needed.  I do like referencing this from the perspective of the VRU (as noted in lines 370-371).  That being said, when pointing out potential issues with CAVs, particularly ADAS or partially autonomous systems, it is really important to discuss the limitation associated with vehicles returning control to the driver under certain conditions.  This could have substantial implications related to the topic covered here as it could certainly impact driver situational awareness, decision making, and perception-reaction time in those instances.

Line 43:  I’m not sure driver expectancy bias is quite as prevalent for a CAV system or would even be an issue.  That being said, if their systems aren’t trained on certain user class behaviors or situations, they could experience deficiencies that could be interpreted similarly.

Line 54:  This is a really good point and relates to the issue with partial autonomous systems referenced prior.

Line 74-75: Around this point I would also discuss the issues/limitations with events or situations that cause the systems to fail and turn control over to the driver (and the implications thereof), per above.

Line 78: I really like discussing the perspective from the VRU – all the questions in lines 81-87 and would be good to study, perhaps involving a survey.

Lines 95-96:  Reasonable approach, need to be careful about this assumption as CAVs have different blind spots versus human-controlled vehicles; for example, one of the biggest issues for human controlled vehicles is an inattentive or distracted driver; this is not as likely of a scenario with a CAV operating in autonomous mode.

Lines 127-128:  I would expect this to indicate more explicitly that dedicated off-street pedestrian facilities may not be present (such as a parallel sidewalk), which would force users onto or closer to the roadway.

Lines 173-174:  One potential perspective here is that the magnitude of the problem that exists, representative of a problem with human-controlled vehicles, offers great opportunity for improvement using ADAS and CAV systems (similar note for Lines 285-287 – i.e., these problems present opportunity).

Lines 185:  This indicates higher likelihood where one would expect interaction and it would be interesting to assess how these systems can account for this in their programming/learning capabilities (which again, aren’t discussed herein).  One potential advantage of a CAV system is that experience can be more readily shared between vehicles.  Much of the ability of a driver to react appropriately is based on personal experience and changes with time behind the wheel.  A CAV system can benefit from the collective learning of the system as widely deployed and tested.  The first CAV operating without a driver in a public setting will utilize many years’ worth of operational data from real world and simulated scenarios.  That’s in stark contrast to the first time a human driver takes to the road without supervision.

Lines 263-265:  This is a very valid observation – roundabouts and other complex intersection or interchange designs (which offer challenges for humans) are a challenge for CAVs.

Lines 302-303: There could be a benefit here as the vehicle would not make the wrong “assumption” based on perception of these cues; instead, they will wait for a safe gap/buffer. A bigger vulnerability here may actually be the conservative action of waiting may increase the likelihood of the vehicle being rear-ended by a human controlled vehicle (and the following driver’s interpretation of the situation).

Line 306:  This again ties into the comparison of human blindspots versus vehicle system blindspots, which may be different.  The noted issue is perhaps less likely with vehicle sensor arrays, though still a potential problem (other blind spots may exist, though likely diminished - still need to be addressed).

Line 320:  Sensor costs are decreasing though.

Lines 346-350:  Really good discussion here and very relevant.

Lines 360-361:  Similar to the issue with motorcycles as previously discussed – less likely with vehicle systems, beginning with ADAS.

Line 365:  As noted above, be careful about constraining the abilities of these systems to perceive commands and possess learning capabilities, though it is still a challenge for sure.

Line 370-371:  Very interesting point here and certainly relevant from the VRU perspective.

Line 389-390:  This is a pretty serious statement and is presented as anecdotal - authors should definitely cite those studies.

Lines 390-396:  There is no evidence to suggest this is not the case – i.e., that perception systems are not being trained using all road users, including motorcycles, bicycles, and pedestrians (they are).  Further, some ADAS features are specifically designed with VRUs in mind (like pedestrian detection alerts).  ADAS and CAV systems are held to industry safety standards and those are publicly available (e.g., ISO and SAE) and could also be referenced.

Author Response

RE: [Behavioral Sciences] Manuscript ID: behavsci-1266237 - Revision

Title: “Autonomous Vehicles and Vulnerable Road-Users – important considerations and requirements based on crash data from two countries”

Dear Reviewers,  

On behalf of myself and my co-authors we would like to say a sincere thank you for your time in reviewing our manuscript, we greatly appreciate your feedback and suggestions for improvement. In addressing your comments, we have made substantial changes to the manuscript for which a revised version is attached, and we have also included below our responses to each comment.

We hope you feel that your comments have been addressed satisfactorily and we look forward to hearing any further response.

With very best regards,

The authors

Reviewer 2

Comment

Response

I believe it would really help if the authors described the types of sensors commonly deployed on these vehicles to support these systems, such as, cameras, LIDAR, and RADAR.  Highlighting their capabilities, as well as the potential blind spots associated with these systems (mentioned at Line 298 but not described or discussed), including how they differ and could be better or worse than human capabilities, or how they could supplement drivers’ perception in the types of situations noted – would provide valuable context.

For example, the authors very adeptly note that vehicles pulling out in front of motorcycles is an issue; however, there isn’t a comprehensive description about how this situation could be mitigated or potentially not, by these systems.  There was one related note about a Netherlands study pointing out that CAVs may not perceive motorcycles as well as other vehicles, but that was the only such example (and it was referenced only, not discussed by the authors in the context of the scenario(s) of interest).  Additionally, one could argue that urban areas and intersections with a lot of activity could be handled better with a full sensor array.  And alternatively, it could be described how or why these sensors may still have trouble due to occlusion, limits in on-vehicle data processing capability, or limitations in the ability to differentiate important information from extraneous input ingested through the perception system, etc.

Additional text has been added to the discussion section to provide context regarding capabilities and limitations of CAV technologies. As this is not the main scope of the current article there is not space to describe this in depth, however additional reference is also made to other research which provide greater detail. 

Abstract Line 25:  I would be a little careful here indicating that CAVs don’t or will not have the capability to detect non-verbal communication.  For instance, there has already been work to get these vehicles to operate under conditions where they are receiving traffic control instructions from a human in the road (such as a police officer). Though certainly more development is needed.  I do like referencing this from the perspective of the VRU (as noted in lines 370-371).  That being said, when pointing out potential issues with CAVs, particularly ADAS or partially autonomous systems, it is really important to discuss the limitation associated with vehicles returning control to the driver under certain conditions.  This could have substantial implications related to the topic covered here as it could certainly impact driver situational awareness, decision making, and perception-reaction time in those instances.

The reviewer makes a good point, and indeed there are rapid developments in this area. Abstract text has been updated, and more added in the main body of the article to more accurately describe the problem that will arise from changing dynamics between VRUs and CAVs, requiring new methods of communication to be learned, and for the VRU to be able to quickly detect what form of communication is required for each specific CAV/human they encounter. References to research in this area have also been added.

Line 43:  I’m not sure driver expectancy bias is quite as prevalent for a CAV system or would even be an issue.  That being said, if their systems aren’t trained on certain user class behaviors or situations, they could experience deficiencies that could be interpreted similarly.

A sentence has been added to clarify that it is not yet clear whether unpredictability will provide the same challenge for CAVs as human drivers, and that appropriate system training will be required.

Line 74-75: Around this point I would also discuss the issues/limitations with events or situations that cause the systems to fail and turn control over to the driver (and the implications thereof), per above.

This is an important area of research, and some text has been added here to mention the issues regarding hand over in partially autonomous vehicles, though again, as this is not the scope of the current paper there is unfortunately not space to go into great detail on the issue.

Lines 127-128:  I would expect this to indicate more explicitly that dedicated off-street pedestrian facilities may not be present (such as a parallel sidewalk), which would force users onto or closer to the roadway.

This has been clarified in the text.

Line 320:  Sensor costs are decreasing though.

This has been clarified in the text.

Lines 360-361:  Similar to the issue with motorcycles as previously discussed – less likely with vehicle systems, beginning with ADAS.

Text has been added earlier in the discussion to give more context around sensors and possible limitations. Reference to that will be made in the text here.           

Line 365:  As noted above, be careful about constraining the abilities of these systems to perceive commands and possess learning capabilities, though it is still a challenge for sure.

This has been clarified in line with earlier additions to the text.

Line 389-390:  This is a pretty serious statement and is presented as anecdotal - authors should definitely cite those studies.

References have been added accordingly

Lines 390-396:  There is no evidence to suggest this is not the case – i.e., that perception systems are not being trained using all road users, including motorcycles, bicycles, and pedestrians (they are).  Further, some ADAS features are specifically designed with VRUs in mind (like pedestrian detection alerts).  ADAS and CAV systems are held to industry safety standards and those are publicly available (e.g., ISO and SAE) and could also be referenced.

Additional text has been added to reflect this, stating that there is indeed development in this area but that more can still be done.

Reviewer 3 Report

This submission deals with an important topic, the safety-related interactions between VRU and ADAS-equipped vehicles. The submission is generally very well written, in parts, but there are several or many typos, misspellings, and other grammatical issues that must be addressed (a few examples are provided below).

  • Authors use “data was” in several places, but they write “data are” on line 289 and “data… suggest” on line 290 – they should decide whether ‘data’ will be considered as singular or plural and remain consistent in its usage throughout
  • Line 289 – typo “althoug”
  • Lines 294 – 300 “Two immediate observations are as folows; (1) that CAVs will need to demonstate a capability to deal with unpredictable actions that VRUs may take at intersections. Examples of these include distracted pedestrians (e.g. using mobile phones, wearing headphones etc.) stepping out into free-flowing traffic and both cyclists and motorcyclists manouevring in vehicle blindspots although many more examples are evident; and (2) that VRUs will need to understand the capabilities and limitaions of CAVs so that the tow roadusers groups can co-exist”
    • While I don’t disagree with either conclusion, I’m not sure that these were derived from the data analyzed in this study – if true, then the value of the analyses conducted is brought into question - this is my biggest challenge with this submission; there seems to be only a tenuous relationship between the data collected/analyzed and the ideas expressed/discussed in the Discussion section.
    • In this one text excerpt, chosen solely based on its content rather than anything to do with its writing, there are several misspelled words and other invented “words” – while generally very well written, the submission requires a thorough review from a skilled technical editor
  • Line 306 – “The 306 challenges of pedestrians appearing from pedestrians crossing…”
    • Authors should consider rewriting this sentence – hard to follow as currently written
  • No concluding paragraph or section – the submission just ends; there is a conclusion 'section' in the abstract; thus, one might expect to see a corresponding section in the article itself

Author Response

RE: [Behavioral Sciences] Manuscript ID: behavsci-1266237 - Revision

Title: “Autonomous Vehicles and Vulnerable Road-Users – important considerations and requirements based on crash data from two countries”

Dear Reviewers,  

On behalf of myself and my co-authors we would like to say a sincere thank you for your time in reviewing our manuscript, we greatly appreciate your feedback and suggestions for improvement. In addressing your comments, we have made substantial changes to the manuscript for which a revised version is attached, and we have also included below our responses to each comment.

We hope you feel that your comments have been addressed satisfactorily and we look forward to hearing any further response.

With very best regards,

The authors

Reviewer 3

Comment

Response

The submission is generally very well written, in parts, but there are several or many typos, misspellings, and other grammatical issues that must be addressed (a few examples are provided below).

The article has now undergone a thorough proofread and the spelling / grammar issues have been addressed.

Authors use “data was” in several places, but they write “data are” on line 289 and “data… suggest” on line 290 – they should decide whether ‘data’ will be considered as singular or plural and remain consistent in its usage throughout

Text has been updated to use plural throughout.

Line 289 – typo “althoug”

This has been corrected.

Lines 294 – 300 “Two immediate observations are as follows; (1) that CAVs will need to demonstrate a capability to deal with unpredictable actions that VRUs may take at intersections. Examples of these include distracted pedestrians (e.g. using mobile phones, wearing headphones etc.) stepping out into free-flowing traffic and both cyclists and motorcyclists manouevring in vehicle blindspots although many more examples are evident; and (2) that VRUs will need to understand the capabilities and limitaions of CAVs so that the tow roadusers groups can co-exist”

           • While I don’t disagree with either conclusion, I’m not sure that these were derived from the data analyzed in this study – if true, then the value of the analyses conducted is brought into question - this is my biggest challenge with this submission; there seems to be only a tenuous relationship between the data collected/analyzed and the ideas expressed/discussed in the Discussion section.

We have tried to address this in the revised Discussion section. We agree with the comment but given the data we used, we cannot explore these 2 issues in detail – instead, we would have to rely on in-depth data which could provide information on road-user behaviour and on-board vehicle technology. The data we have used is macroscopic in nature and cannot give the level of detail to adequately address the questions. This could be considered to be the basis of a future study. 

         • In this one text excerpt, chosen solely based on its content rather than anything to do with its writing, there are several misspelled words and other invented “words” – while generally very well written, the submission requires a thorough review from a skilled technical editor

The article has now undergone a thorough proofread and the spelling / grammar issues have been addressed.

Line 306 – “The 306 challenges of pedestrians appearing from pedestrians crossing…”

Authors should consider rewriting this sentence – hard to follow as currently written

The authors agree the wording here was confusing, the sentence has been rewritten.

No concluding paragraph or section – the submission just ends; there is a conclusion 'section' in the abstract; thus, one might expect to see a corresponding section in the article itself

A Conclusions section has now been added.

Reviewer 4 Report

The article is interesting; however, I believe that some changes must be made before it can be published. In the introduction, the situation of Great Britain and Australia in what concerns the topic of the study could be explained in order to provide readers with more context. In addition, more emphasis could be put on the importance of the attitudes of the population regarding the acceptance of users and the use intention. Therefore, I recommend the following articles: Montoro, L., Useche, S. A., Alonso, F., Lijarcio, I., Bosó-Seguí, P., & Martí-Belda, A. (2019). Perceived safety and attributed value as predictors of the intention to use autonomous vehicles: A national study with Spanish drivers. Safety Science, 120, 865-876; Pettigrew, S., Worrall, C., Talati, Z., Fritschi, L., & Norman, R. (2019). Dimensions of attitudes to autonomous vehicles. Urban, Planning and Transport Research.; y, Lijarcio, I., Useche, S. A., Llamazares, J., & Montoro, L. (2019). Perceived benefits and constraints in vehicle automation: Data to assess the relationship between driver's features and their attitudes towards autonomous vehicles. Data in brief, 27, 104662.

In addition, the objective of the research must be clearly explained. In the method, I suggest explaining in detail elements such as the statistical package that was used for the data analysis, how the data will be processed, and the ethical aspects of the study. The results are adequately explained. And, even though the descriptive data provide relevant information, it is possible that, considering the database the authors had access to, more analyses could have been performed (for instance, ANOVA or others).

Generally speaking, the articles has few references, reason why I suggest including new ideas in the discussion, with their respective references, in order to contrast the obtained results. For instance, research referring to the interaction of autonomous vehicles with different road users and their possible risky behaviors could be included. Or else, providing an approximation on the predisposition of people towards new technologies in traffic, which could have an impact and show a relation with their intention of using autonomous vehicles, as well as their attitude towards them. Along this line, I recommend the following article, which explains this circumstance in depth: Alonso, F., Faus, M., Esteban, C., & Useche, S. A. (2021). Is There a Predisposition towards the Use of New Technologies within the Traffic Field of Emerging Countries? The Case of the Dominican Republic. Electronics, 10(10), 1208.

Lastly, I suggest including a brief conclusion section, as well as the limitations of the study.

Watch out for typos, for instance in the abstract: “Crash data were analyses…”, you meant “analysed”.

Author Response

RE: [Behavioral Sciences] Manuscript ID: behavsci-1266237 - Revision

Title: “Autonomous Vehicles and Vulnerable Road-Users – important considerations and requirements based on crash data from two countries”

Dear Reviewers,  

On behalf of myself and my co-authors we would like to say a sincere thank you for your time in reviewing our manuscript, we greatly appreciate your feedback and suggestions for improvement. In addressing your comments, we have made substantial changes to the manuscript for which a revised version is attached, and we have also included below our responses to each comment.

We hope you feel that your comments have been addressed satisfactorily and we look forward to hearing any further response.

With very best regards,

The authors

Reviewer 4

Comment

Response

In the introduction, the situation of Great Britain and Australia in what concerns the topic of the study could be explained in order to provide readers with more context. In addition, more emphasis could be put on the importance of the attitudes of the population regarding the acceptance of users and the use intention. Therefore, I recommend the following articles: Montoro, L., Useche, S. A., Alonso, F., Lijarcio, I., Bosó-Seguí, P., & Martí-Belda, A. (2019). Perceived safety and attributed value as predictors of the intention to use autonomous vehicles: A national study with Spanish drivers. Safety Science, 120, 865-876; Pettigrew, S., Worrall, C., Talati, Z., Fritschi, L., & Norman, R. (2019). Dimensions of attitudes to autonomous vehicles. Urban, Planning and Transport Research.; y, Lijarcio, I., Useche, S. A., Llamazares, J., & Montoro, L. (2019). Perceived benefits and constraints in vehicle automation: Data to assess the relationship between driver's features and their attitudes towards autonomous vehicles. Data in brief, 27, 104662.

Additional text and references have been added to the introduction to give context regarding user trust and acceptance of CAVs.

In addition, the objective of the research must be clearly explained. In the method, I suggest explaining in detail elements such as the statistical package that was used for the data analysis, how the data will be processed, and the ethical aspects of the study. The results are adequately explained. And, even though the descriptive data provide relevant information, it is possible that, considering the database the authors had access to, more analyses could have been performed (for instance, ANOVA or others).

This is addressed in the revised version of the paper.

Due to the different nature of data collection in the 2 countries it was suggested that statistical analysis of the data could provide mis-leading outputs. In the GB data analysis, the data are representative of the GB injury crash population due to the nature of data collection. In Australia, we could not make the same assumptions as data were only available in 2 States. Whilst we think that reasonable representativity is evident, we cannot be 100% confident and therefore we considered it best to use more descriptive analysis rather than our initial intention to use ANOVAs.

Generally speaking, the articles has few references, reason why I suggest including new ideas in the discussion, with their respective references, in order to contrast the obtained results. For instance, research referring to the interaction of autonomous vehicles with different road users and their possible risky behaviors could be included. Or else, providing an approximation on the predisposition of people towards new technologies in traffic, which could have an impact and show a relation with their intention of using autonomous vehicles, as well as their attitude towards them. Along this line, I recommend the following article, which explains this circumstance in depth: Alonso, F., Faus, M., Esteban, C., & Useche, S. A. (2021). Is There a Predisposition towards the Use of New Technologies within the Traffic Field of Emerging Countries? The Case of the Dominican Republic. Electronics, 10(10), 1208.

Additional references have been added regarding user trust and acceptance, methods that CAVs and VRUs may use to communicate, and regarding sensor technologies and limitations.

The reference given here has been added in the context of suggesting that the current study is limited to 2 developed countries, and future work should consider attitudes and behaviours of road-users in less developed countries.

Lastly, I suggest including a brief conclusion section, as well as the limitations of the study.

This is addressed in the paper.

Watch out for typos, for instance in the abstract: “Crash data were analyses…”, you meant “analysed”.

This has been corrected, and a more thorough proof-read of the entire article has been completed

Round 2

Reviewer 2 Report

The authors did a very good job improving the context of the paper, providing additional references and discussion points that are very relevant to the topic of the paper.  Further, I hope through the additional review and revision, the authors gained additional perspective on where future research should be directed in this critical area of study.  In all, I appreciate the effort to improve the paper based on the feedback provided.

I have two very minor comments I recommend prior to publication (though optional):

Line 475 - suggest saying the "...sensors will need to have enhanced capability..." as "a wider range" doesn't quite capture what I think you are after here, unless you are explicitly talking about their range of detection (as in, how far away they can correctly perform object detection).

Conclusions:  Bullet 5 - suggest also reiterating here the issue with handover control in partially autonomous vehicles where certain events dictate the need (which could involve intersections/VRUs).

Reviewer 4 Report

The authors have taken into account the suggestions I provided in my previous review, so I consider that the manuscript is suitable for publication